# Group-format, peer-facilitated mental health promotion interventions for students in higher education settings: a scoping review protocol

Carrie Brooke-Sumner ,[1,2] Mercilene T Machisa ,[2,3] Yandisa Sikweyiya,[3,4] Pinky Mahlangu[2,3]

¹Mental Health, Alcohol, Substance Use and Tobacco Research Unit, South African Medical Research Council, Cape Town, South Africa
²School of Nursing and Public Health, College of Health Sciences, Howard College Campus, University of KwaZulu Natal, Durban, South Africa
³Gender and Health Research Unit, South African Medical Research Council, Pretoria, South Africa
⁴School of Public Health, University of the Witwatersrand, Johannesburg, South Africa

**Correspondence to**
Dr Carrie Brooke-Sumner; carrie.brooke-sumner@mrc.ac.za

## ABSTRACT

**Introduction** Young people in higher education face various stressors that can make them vulnerable to mental ill-health. Mental health promotion in this group therefore has important potential benefits. Peer-facilitated and group-format interventions may be feasible and sustainable. The scoping review outlined in this protocol aims to map the literature on group-format, peer-facilitated, in-person interventions for mental health promotion for higher education students attending courses on campuses in high and low/middle-income countries.

**Methods and analysis** Relevant studies will be identified through conducting searches of electronic databases, including Medline, CINAHL, Scopus, ERIC and PsycINFO. Searches will be conducted using Boolean operators (AND, OR, NOT) and truncation functions appropriate for each database. We will include a grey literature search. We will include articles from student participants of any gender, and published in peer-reviewed journals between 2008 and 2023. We will include English-language studies and all study types including randomised controlled trials, pilot studies and descriptive studies of intervention development. A draft charting table has been developed, which includes the fields: author, publication date, country/countries, aims, population and sample size, demographics, methods, intervention type, comparisons, peer training, number of sessions/duration of intervention, outcomes and details of measures.

**Ethics and dissemination** No primary data will be collected from research participants to produce this review so ethics committee approval is not required. All data will be collated from published peer-reviewed studies already in the public domain. We will publish the review in an open-access, peer-reviewed journal accessible to researchers in low/middle-income countries. This protocol is registered on Open Science Framework (https://osf.io/agbfj/).

## STRENGTHS AND LIMITATIONS OF THIS STUDY

⇒ This scoping review will map literature on group-format, peer-facilitated and in-person interventions for mental health promotion among higher education students globally.
⇒ The methods are grounded in established methods and guidance for scoping reviews (Preferred Reporting Items for Systematic Reviews and Meta-Analyses).
⇒ The review will include a search of published and grey literature to synthesise the range of interventions for group-based, in-person, peer-facilitated mental health promotion for students.
⇒ The protocol is limited in focusing on English-language articles.

## BACKGROUND

Mental health of young people is a global public health priority,[1] with COVID-19 having accentuated the urgency of addressing this area.[2 3] Mental well-being is crucial for higher education (university/college) students to be resilient in facing the demands of academic life[4] as well as being a building block for every society through youth development. Young people entering higher education face a period of substantial change, with new and varied stressors and changing interpersonal relationships, financial circumstances and life challenges.[5–7] Academic and other stresses are associated with the development of mental health conditions.[5 7] Mental distress is thus prevalent in higher education institutions with young female students particularly vulnerable to conditions such as anxiety, depression and post-traumatic stress disorder (eg, 8–11). The stressors experienced in student life overlay other vulnerability factors and social determinants[12] for developing a mental health condition, including previous experiences of childhood adversity and trauma,[13 14] and being from a sexual minority group or gender nonconforming.[13] In low/middle-income countries (LMICs) in which human resources for mental health treatments are especially scarce,[15] the potential gains from prevention of mental disorders for young people

are significant and may correspond to population-level benefits in the long term.[16 17]

Mental health promotion (encompassing building well-being and prevention of development of mental health conditions) remains underprioritised in many settings in comparison with efforts for treatment provision. However, such work is a crucial component of the WHO mix of mental health services (which includes self-care and informal care, primary healthcare and specialist healthcare).[18] Good mental health for young people may be contextually nuanced, but is dependent on a range of domains, including mental health literacy, attitudes towards mental health conditions, self-perceptions, cognitive skills, academic performance, emotions, behaviours, self-management, social skills, quality of significant relationships, physical and sexual health, meaning in life and quality of life.[19 20] With this range of domains, transdiagnostic or common elements of treatment and prevention approaches are relevant,[17] underpinned conceptually by the building of adaptive social and emotional skills, stress management, positive self-perceptions and supportive relationships. Prevention in relation to mental ill-health also encompasses other health behaviours (eg, dietary health, physical activity, sleep) that can have an impact on mental health outcomes.[21] Harnessing common practice elements can enable an evidence-informed approach to prevention even when a specific evidence-based intervention may be lacking (as may be the case in many LMIC settings where this research area is less developed).[17 22]

Given the breadth of domains of good mental health for higher education students, mental health promotion and prevention interventions necessarily cover a wide variety of interventions at the individual, community and societal levels[23] which can be universal (population based) or selective (targeting individuals or subgroups at higher risk).[16 24] Young people in higher education settings present a particular set of opportunities for preventive and promotive interventions given the social connectedness of student populations. Many mental health conditions have their onset in adolescence. This is a key point for intervention and although not the focus of this review, there is developing evidence that school-based prevention can be effective in improving mental health literacy and reducing mental health stigma.[25–28]

There is also a developing systematised evidence base for higher education student mental health promotion from high-income countries (HICs). Systematic reviews of prevention programmes for student mental health showed moderate effects for common practice elements including psychoeducation (mental health literacy training), relaxation techniques and cognitive restructuring.[19] Guided mental health skills training programmes[22 29–31] and computer and web-based interventions delivered by a variety of professionals[32–34] similarly showed moderate effects. Current developments in mental health promotion include a movement towards a positive conceptualisation of mental health literacy which is not focused on 'illness' and symptoms. Within this conceptualisation, positive mental health literacy includes problem-solving ability, independence, relational skills and self-control.[35] A similar evidence base for student mental health promotion in LMICs is required[36] building on evidence for approaches from HICs, but with consideration for contextual differences in relation to availability of human and other resources.

Most notably, a key aspect of mental health prevention interventions is the delivery agent. In LMIC settings, specialists such as psychologists, social workers and counsellors are scarce[37] and task-sharing (service provision by non-specialists with upward referral if necessary) has long been advocated as a feasible approach for mental health service delivery.[23 38 39] Digital interventions have also been put forward as a low-resource approach for mental health promotion.[36] Evidence for digital interventions for mental health and access to technology in HICs and LMICs is growing and indicates potential for these approaches, in conjunction with other supportive service options.[40 41] As such, in-person interventions enable building of rapport and support and are needed in contexts where access to computer or mobile technology may be limited. Group approaches may be a feasible low-resource approach in some LMIC contexts. Hence, in-person group approaches are the focus of this review, which will inform development of in-person group interventions. The group format presents the opportunity for reaching higher numbers of students in a less resource-intensive approach than individual-level support, which may be advantageous for mental health promotion.[36] Further to this, peer-led approaches, in which people who share a common lived experience (eg, being a higher education student) are involved in providing services, have traction in other areas of public health, for young people in particular (eg, [42 43]) and in other areas of mental health service provision (eg, [44 45]). Peer-facilitated approaches may be particularly appropriate for mental health promotion among young people given the importance of peer influence at this point in the life course.[46] These approaches have the potential to be low resource and sustainable in resource-constrained settings,[42] though still requiring investment and engagement[47] to be productively implemented.

This scoping review will map the evidence for in-person, group-based, peer-facilitated mental health promotion interventions for young people in higher education settings globally. Once conducted, this review will enable progress in bringing together evidence for such preventive approaches that are applicable for resource-constrained LMIC contexts. This will serve as a step towards building the field of evidence-informed mental health promotion among students in higher education settings in LMICs.

### Research questions

1. What range of in-person, group-based, peer-facilitated mental health promotion interventions are offered to higher education students on campuses globally?
2. What are the common practices, processes and outcome measures used for these interventions?

## Objectives

1. To map the literature on group-based, peer-facilitated, in-person group interventions for mental health promotion for higher education students on campuses globally.
2. To describe these peer-facilitated group interventions and the ways they have been evaluated.
3. To describe common practice elements and those identified as having positive effects, no effects or negative effects, if possible.

## METHODS AND ANALYSIS

Methodology for the review will be based on that presented by Arksey and O'Malley[48] and further developed by Levac et al.[49] A key recommendation from this advancement relates to clarifying and linking the purpose of the review with the research question.[49] The purpose of the review will be to identify common practice elements of group-format, peer-facilitated mental health promotion interventions used in higher education populations. We will include a grey literature search as recommended by Joanna Briggs Institute Guidance.[50 51] We will also use an iterative approach for identifying and selecting studies[49] including identifying studies from reference lists of relevant studies with discussion of the review team. The following steps will be undertaken to investigate the review question and are detailed in the sections that follow:

1. Identifying the research question
2. Identifying relevant studies
3. Study selection
4. Charting the data
5. Collating, summarising and reporting results

### Identifying the research question

The Population, Concept, Context mnemonic has been identified as appropriate for scoping review methodology.[50] The population of interest is defined as students studying at higher education campuses; the concept is defined as any mental health promotion intervention delivered in person by a peer and in a group format; context is defined as articles from HICs and LMICs. This scoping review therefore has the following research questions:

#### Primary research question

► What evidence is available globally for peer-facilitated, group-format in-person interventions for mental health promotion for students on higher education campuses?

#### Secondary research questions

► What are the common practice elements and delivery methods for interventions reported in this body of literature?
► What research gaps are identified by the literature included in the scoping review?

### Identifying relevant studies

The review will be conducted from June to December 2024. Relevant studies will be identified through conducting searches of electronic databases, including Medline, CINAHL, Scopus, PsycINFO and ERIC. Searches will be conducted using Boolean operators (AND, OR, NOT) and truncation functions appropriate for each database. Search strings will be developed in line with literature on peer approaches in higher education[52] and according to each database. Searches will include for example: "mental health literacy" OR "mental health promotion" OR "psychoeducation" AND "intervention" OR "program" OR "mental health intervention" AND "group intervention" OR "peer" OR "peer support" OR "peer health education" AND "student" OR "campus" OR "university" OR "college" AND "depression" OR "anxiety" OR "mental health literacy" OR "positive mental health" OR "self care" OR "resilience" or "stress management". We will seek assistance from a specialist librarian in refining search strings for databases based on these key terms to generate search results feasible for review by the study team. We will take an iterative approach to the searches[50] to include relevant keywords and to maintain feasibility of the number of records to review. Review authors will conduct backward citation tracking for all articles selected for inclusion as an approach to identify further relevant studies. If required, review authors will contact corresponding authors of these articles to identify relevant studies.[50] Relevant studies will be downloaded and managed using Rayyan software to conduct screening. The grey literature search will be conducted with terms above using databases such as WorldCat, OpenMD, Eldis, OpenGrey, Canadian Agency for Drugs and Technologies in Health and Grey Matters. Programme reports, case studies and manuals will be included.

### Study selection

Titles and abstracts of identified studies will be subjected to double-screening for relevance, with records divided among authors to facilitate faster screening. Titles and abstracts will be screened according to the predeveloped inclusion and exclusion criteria developed for answering the review question (figure 1). Articles identified as relevant in this first screening process will then be included in a full-text review conducted by all authors (double-screening, two authors review each article). Agreement and disagreement on full-text inclusion will be discussed by reviewing authors, and any disagreements resolved through discussion with the full review team.

### Inclusion criteria

The purposes of the scoping review mean that included articles will describe in-person interventions delivered on higher education campuses in HICs and LMICs, by peers. Articles from HICs will be included as there may be practice elements and delivery approaches from HICs that are relevant for advancing the field in LMICs where peer-delivered approaches may be less well developed.

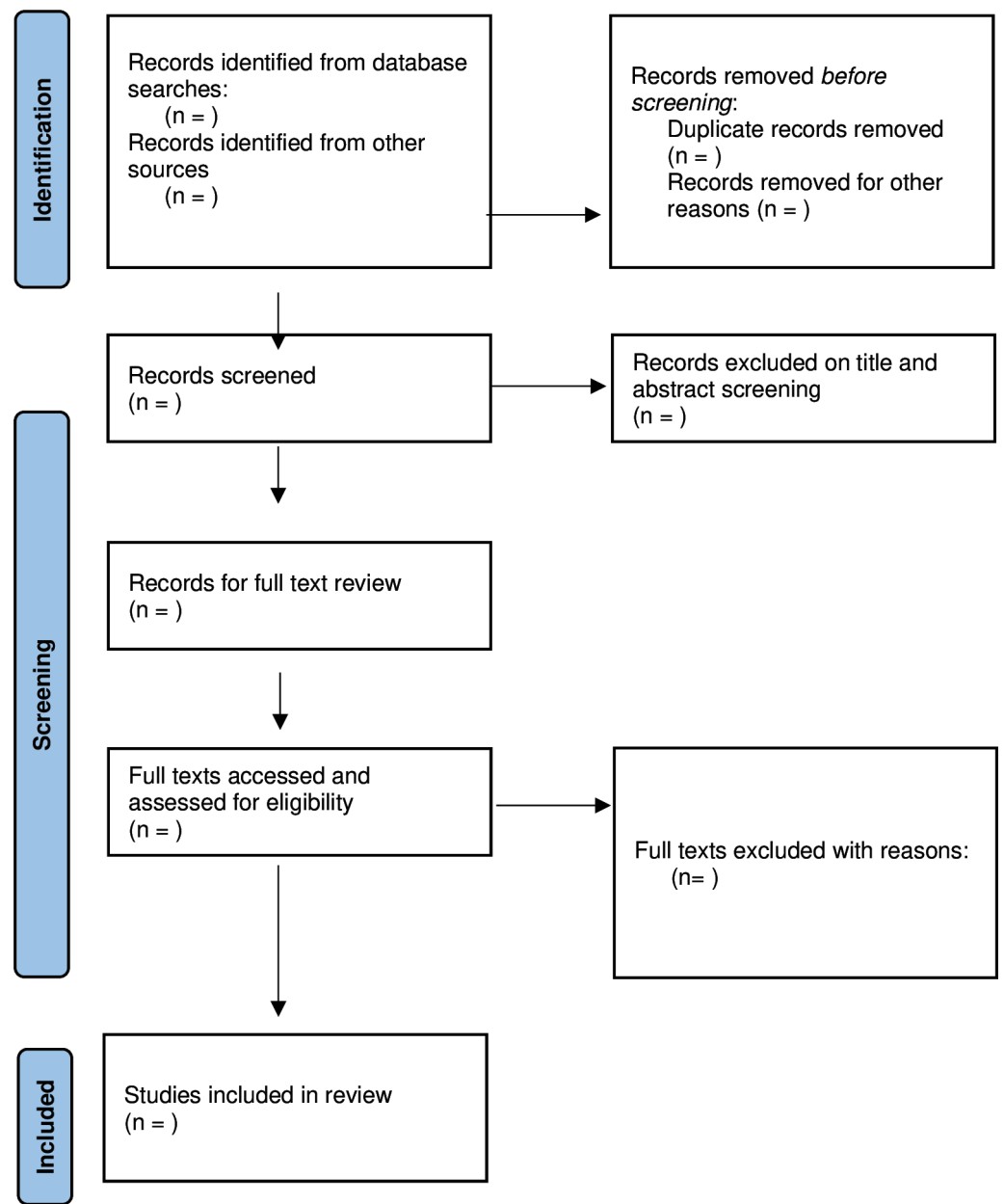

**Figure 1** Preferred Reporting Items for Systematic Reviews and Meta-Analyses flow diagram describing the process of article review and inclusion and exclusion for the scoping review.

We will include articles that describe group-format interventions delivered in higher education settings among student participants of any gender, and published in peer-reviewed journals between 2008 and 2024 (15-year range for feasibility in number of records to review and to capture recent developments in the field). We will include English-language studies and all study types including randomised controlled trials, pilot studies and descriptive studies of intervention development. We will not exclude papers that do not report where the study was conducted.

### Exclusion criteria
Articles describing individual and group interventions delivered by clinical professionals (eg, psychologists, social workers, counsellors or other university staff) will be excluded, as well as articles reporting on clinical treatment interventions as opposed to prevention interventions. Articles reporting on interventions not delivered in person, by peers or in groups (eg, online/web based) will also be excluded. Non-English-language papers and those published outside of the time period indicated will be excluded.

### Data extraction and management
Data extraction will be conducted from PDF or other electronic files into an Excel-based data management file (online supplemental file 1). We will conduct a pilot test of this abstraction form with initial articles to ensure consistency in data extraction between team members.

### Charting the data
The charting process will provide a descriptive summary of the evidence presented in included studies,[50] as the

initial step towards answering the primary and secondary research questions. The draft charting table includes the fields: author, publication date, country/countries, aims, population and sample size, methods, demographics of peers and/or target population, intervention type, peer training, number of sessions/duration, retention, comparisons, duration of intervention, outcomes and details of measures. The data charting table completed for each reviewer will be collated and reviewed by the whole review team to address any inconsistencies.

### Assessment of study quality

For this scoping review, we will not conduct a formal assessment of risk of bias using one of the established reporting tools (eg, [53] [54]) as the aim is to provide an overview of existing evidence regardless of methodological quality. We will however use the Template for Intervention Description and Replication (TIDieR)[55] checklist to assess completeness in the reporting of included intervention studies. Available since 2014, this guide may have been used by study authors to describe interventions and can be used in this scoping review to provide a characterisation of interventions along with charting of data.

### Summarising and reporting findings

Appropriate tables will be used to give the reader an overall picture of the spread of evidence across publication years and country of origin. A narrative description of the studies will also be presented (including TIDieR), aligning findings with the primary and secondary research questions.[50] For ease of use for the reader, findings will be presented under the conceptual headings 'Definition of peer', 'Intervention practice elements', 'Intervention delivery' and 'Research gaps'. The final manuscript will be prepared using the Preferred Reporting Items for Systematic Reviews and Meta-Analyses extension for scoping reviews.[56] Depending on the amount of literature found, data abstraction may also include categorising the extracted information into themes to identify patterns across the literature, and according to high-income, low-income or lower middle-income country status (given differences in availability of resources across these settings).

### Ethical considerations and dissemination

No primary data will be collected from research participants to produce this review. All data will be collated from published peer-reviewed studies already in the public domain. This study therefore does not require ethical approval from a Human Research Ethics Committee. To minimise potential bias and ensure an accurate reflection of the scope of the literature, we have a review team with four members who will work collaboratively. These researchers are senior scientists at the South African Medical Research Council, with backgrounds in quantitative and qualitative research in public health and mental health. We will publish the review in an open-access, peer-reviewed journal accessible to researchers in LMICs. We will also disseminate findings through our established Community Advisory Board and stakeholder engagement platforms of the South African Medical Research Council.

### Patient and public involvement

Patients and/or the public were not involved in the design, or conduct, or reporting, or dissemination plans of this research.

### DISCUSSION

The United Nations Sustainable Development Goals (SDGs) prioritise mental wellness in SDG 3.[57] This indicates high-level commitment to improvement in mental well-being, which requires a response from all sectors of society, and quality research to guide this response.[58] Addressing social determinants of mental health,[12] the social context that precedes the development of mental health conditions (eg, through mental health promotion) holds great potential for prevention of the growing burden of mental disorders globally and in young people in particular who make up a large proportion of the global population. Young people in higher education in particular face developmental and academic stressors which can make them vulnerable to mental health conditions.[59–61]

Prevention efforts that promote mental health will need to begin to address vulnerability as well as improve coping and resilience. Intervention research to develop and test interventions in this area is required.[58] The scoping review outlined in this protocol will contribute to moving the mental health promotion field in LMICs forward through mapping group-format, peer-facilitated mental health promotion practice elements. This will add to the body of work on adaptation of complex interventions,[62] [63] incorporating learning from HIC settings with practice elements that are appropriate and can be tailored for LMIC settings. Evidence for approaches that are feasible, acceptable and crucially tailored for LMIC contexts is required, given the risk factors associated with mental health conditions in these settings.[60] [64] For example, such approaches may give prominence to developing social support, promoting aspects of well-being including social, academic and spiritual well-being, and building resilience and coping strategies which are crucial in LMIC contexts.[65] Through mapping the body of literature, this scoping review will also contribute to evidence for state of the field of task-sharing for mental health promotion in LMIC settings. Finally, findings from the review will identify gaps and opportunities for researchers and practitioners interested in implementing group-format, peer-led approaches for improving mental health for students in LMICs and in HICs.

**Contributors** All authors conceptualised the study and developed the methods collaboratively. PM, MTM and YS contributed to conceptualisation of research questions, development of search strategy and scoping review methods. CB-S led the development of the research questions and methods and produced the initial draft of this manuscript. All authors reviewed and approved the final manuscript.

**Funding** CB-S, MTM, PM and YS are supported in this work by funds from the South African Medical Research Council Flagship projects funding.

**Competing interests** None declared.

**Patient and public involvement** Patients and/or the public were not involved in the design, or conduct, or reporting, or dissemination plans of this research.

**Patient consent for publication** Not applicable.

**Provenance and peer review** Not commissioned; externally peer reviewed.

**ORCID iDs**
Carrie Brooke-Sumner http://orcid.org/0000-0002-9489-8717
Mercilene T Machisa http://orcid.org/0000-0001-7275-1100

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
