## [Reviewer comments · BMJ Open]

ARTICLE DETAILS

TITLE (PROVISIONAL)	Scoping review protocol for group format, peer-facilitated mental health promotion interventions for students in higher education settings
AUTHORS	Brooke-Sumner, Carrie; Machisa, Mercilene T.; Sikweyiya, Yandisa; Mahlangu, Pinky

VERSION 1 – REVIEW

REVIEWER	Judith Lunn Lancaster University
REVIEW RETURNED	07-Dec-2023

GENERAL COMMENTS	The proposed study protocol aims to review interventions that use peer learning or group work to address mental health and wellbeing in tertiary sector students in Low-Middle Income Countries (LMICs). An interesting and important topic. The protocol is clear in terms of concepts, context and population. The search terms could be more specific to hone in on relevant literature. There are a few other comments and questions outlined below. Abstract Clear and concise statement of background and aims, context and population. Is being on a campus in a LMIC a requirement for inclusion, as this suggests the need to include information as to whether participants of studies live on campus versus other off campus at other locations. Methods and Analysis Clear and precise with all relevant information present. State the minimum number in the review team. Strengths and Limitations If word count allows, state the name of the guidance followed. There is also value here for those both inside and outside of LMICs who wish to address student mental health. This could potentially be better emphasised here. As well as why focussing exclusively on the LMIC context is important. Keywords There is no mention of LMICs although this is the specific context of focus.
--

	Background: Last sentence in first paragraph is difficult to grasp and would benefit from rewording. There is a difference between asymptomatic and sub-clinical or mild to moderate, unclear what is meant here. WHO needs to be spelled out in full at first mention and mix of services needs to be described, albeit briefly. The term 'resilience' is not without controversy, it may be worth checking this. Line 85 missing a close bracket. Need to clarify if any difference between school-based and tertiary education contexts of interventions and is unclear from this if school-based studies will be included. Objective 3. It may be preferable to state that common practice elements will be described in full, and then those identified as having positive effects, no effects, negative effects, if possible. The population may not always be 18-30 so maybe not helpful to be so specific in terms of age. Identifying search terms What about 'psychoeducation'? There is no mention of LMIC in searches that may result in exceptionally high number of returns. It is recommended a librarian is contacted as part of this review process for assistance on how to develop search strategy to capture those studies. There is greater specificity in terms of mental health literacy (self-care, resilience etc) but not in terms of mental health outcomes (anxiety, depression etc). If it is a small pool of studies further extending to these concepts may capture more of the interventions you wish to review. There could be more details provided about inclusion criteria for types of grey literature. There needs to be clearer emphasis earlier in the manuscript that you are only interested in in-person delivery and why. It is mentioned above but just needs to be stated more clearly. Make a statement as to why formal assessment bias is not appropriate for this context. TIDier is the appropriate tool. Will the summarising of findings be a reiterative process? Discussion Social determinants are mentioned but not part of the scoping protocol or research objectives. Will variation in country of origin take into account relevant social deprivation indices, for example. This may have relevance to implementation of any key recommendations and findings.
REVIEWER	Julia Pointon-Haas King's College London, Psychology

GENERAL COMMENTS

Dear authors,

Thank you for the opportunity to read your protocol! This is an exciting scoping review that I look forward to reading in the future. I hope that my comments are helpful.

Best wishes with the review,

Julia

Below are my comments:

1) What is mental health promotion? Perhaps add a definition of mental health promotion in paragraph starting at line 75.

2) Line 85: missing bracket after references 23, 27, 28

3) Higher education or tertiary education or university – in UK, these all mean the same thing, but in the United States, there are more nuanced definitions. Clarify what you mean by the terms at the start and outline which wording you'll use throughout, or just use one. Either way, define the setting.

4) This is a very long sentence to read. Perhaps break up so that it doesn't lose the important points it is making: Given that many mental 98 health conditions have their onset in adolescence this is a key point for intervention and there is 99 developing evidence that school-based prevention programmes can be effective in improving 100 mental health literacy and reducing mental health stigma defined as attitudes and beliefs regarding 101 mental disorders [25-28].

5) Split this long sentence into two for strong reading and so that the paragraph has at least 3 sentences: Systematic reviews of prevention programmes for student 105 mental health showed moderate effects for common practice elements including psychoeducation, 106 relaxation techniques, and cognitive restructuring [19] as well as guided mental health skills 107 training programmes [22, 29-31] and computer and web-based interventions delivered by a variety 108 of professionals [32-34]

6) Paragraph starting from line 111: Why have you chosen to focus on psychoeducation when you've listed other prevention programmes (i.e. relaxation techniques, cognitive restructuring, etc.) in the paragraph before that showed moderate effects?

7) Line 131, it may be useful to state why in-person interventions 'contribute to a different niche'

8) Line 141, 'These approaches have the potential to be low resource....' Some research argues that peer-led approaches are not 'cheap' (Turner, G. Peer support and young people's health. *J. Adolesc.* 1999, 22, 567–572.) I found something similar when speaking to staff who run peer programmes in higher education settings. Staff required more resources and capacity than they had to run the programmes effectively, which you may want address briefly. See my recent publication: Pointon-Haas J, Byrom N, Foster J, Hayes C, Oates J. Staff Perspectives: Defining the Types, Challenges and Lessons Learnt of University Peer Support for Student Mental Health and Wellbeing. *Education Sciences.* 2023; 13(9):962. <https://doi.org/10.3390/educsci13090962>

9) Line 145: 'in education and training settings' – once again, clarify settings at start, this is first time you've used term 'training settings.'

10) Research question 1: add 'in-person' as you have outlined in line 144. Also, with terms, in-person, group-based and peer-facilitated, use the hyphen (or don't) consistently.

11) Line 186: The chosen age range (18-30) may limit a full picture for this review, as tertiary education students can be outside this age range, while I agree that the majority are not. Also, some studies

	may not report on age, so will you exclude these? Or, their age range may not match yours, so will you exclude these? It may be easier to just say 'tertiary education students' and remove the age range altogether. 12) Line 201: you may also want to consider using the educational database, ERIC (Education Resources Information Center) 13) Line 211: you can use term 'backward citation tracking' to describe looking at reference list of included studies. 14) Line 217: perhaps consider using OpenGrey and Grey Matters and cite the grey literature searches. 15) For searching, perhaps you can use existing relevant research to help you define your search terms. For example, in my recent systematic review, I used: John NM, Page O, Martin SC, Whittaker P. Impact of peer support on student mental wellbeing: a systematic review. MedEdPublish 2018; 7(3): 170. The search strategy I developed in my recent systematic review may help you decide an exhaustive search term list: Pointon-Haas J, Waqar L, Upsher R, Foster J, Byrom N, Oates J. A systematic review of peer support interventions for student mental health and well-being in higher education. BJPsych Open. 2023 Dec 15;10(1):e12. doi: 10.1192/bjo.2023.603. PMID: 38098123; PMCID: PMC10755562. 16) Line 218: while it's for systematic reviews to do independent screening, which may not be relevant for the scoping review, I found Rayyan helpful to use for screening after downloading citations into Endnote in a recent review: https://www.rayyan.ai/ 17) Line 227, Inclusion Criteria: you may want to say why you have chosen the range of years '2007 to 2023' and also say when you are planning to undertake this search. Also, the impression I had reading this was that the review would only be of LMIC campuses, but you've included HIC as well in the inclusion criteria. Perhaps clarify this. 18) Line 228: I'm assuming that you'll include any study internationally? Perhaps it would be useful to clarify that this is a global scoping review somewhere. Also, some studies do not report on the country where the study took place, I assume you will exclude those? It may be good to elucidate this in exclusion criteria. 19) Line 243, I think you're missing a word? 'Data extraction from...will be' 20) Line 250: perhaps it would be good to add how the study defines a 'peer' into your fields. How a 'peer' is defined beyond being a student is interesting to see, since clinical settings define peer support workers as people with lived experience of mental health difficulties who use those experiences to share and help others. In contrast, tertiary education settings tend to bring students together based on the courses they share or their year of study. It may be interesting to gather this data to see how it is defined in studies. 21) Line 250: perhaps consider adding the number of sessions that took place in addition to the duration of the intervention. Many peer programmes have a set number of sessions that take place, and they don't report on duration. 22) Line 250: In addition to sample size, you may want to report on retention if the programme happens over a period of time. Also, will you report on demographics of the peer leading the sessions and/or those attending the peer group if provided? 23) Line 250: How will you report on the 'positive effects on mental health' as your 3rd research objective outlines you will do? Will there be any sort of meta-analysis of statistics presented to consider effectiveness? In the 'summarizing and reporting findings' section, you may want to outline that you don't expect to be able to do a meta-analysis as the measures used have been reported to be
--	--

	heterogenous (see Pointon-Haas systematic review provided in point 15). 24) Line 250: perhaps you also want to add details on the training that the peers leading programme received, or the professional support they were given. These are the areas where costs are incurred, which make peer programmes less economical than sometimes assumed. (see point 8) 25) Line 258: how will you report on your assessment of completeness? Will this be charted with the other data? 26) Line 292: it may also be useful to say how big the population of students in tertiary education are to give an idea of scope.
--	---

VERSION 1 – AUTHOR RESPONSE

Reviewer 1 Comment	Author response
The proposed study protocol aims to review interventions that use peer learning or group work to address mental health and wellbeing in tertiary sector students in Low-Middle Income Countries (LMICs). An interesting and important topic. The protocol is clear in terms of concepts, context and population. The search terms could be more specific to hone in on relevant literature. There are a few other comments and questions outlined below. Abstract Clear and concise statement of background and aims, context and population. Is being on a campus in a LMIC a requirement for inclusion, as this suggests the need to include information as to whether participants of studies live on campus versus other off campus at other locations.	Thank you for this comment, the search terms have been made more specific as per Reviewer 2 comments also. Thank you for this comment. We have added ‘attending courses’ line 28, to indicate that it could be students who live on campus or who live elsewhere but who are attending courses on the campus.
Methods and Analysis Clear and precise with all relevant information present. State the minimum number in the review team.	

	Thank you, added to line 41.
Strengths and Limitations If word count allows, state the name of the guidance followed. There is also value here for those both inside and outside of LMICs who wish to address student mental health. This could potentially be better emphasized here. As well as why focussing exclusively on the LMIC context is important.	PRISMA guidance added, line 50. Thank you, this point has now been added, line 48
Keywords There is no mention of LMICs although this is the specific context of focus.	In line with reviewer 2 comments, we have clarified that the review will include HIC and LMIC studies, so this is still not included in the keywords.
Background: Last sentence in first paragraph is difficult to grasp and would benefit from rewording. There is a difference between asymptomatic and sub-clinical or mild to moderate, unclear what is meant here. WHO needs to be spelled out in full at first mention and mix of services needs to be described, albeit briefly. The term 'resilience' is not without controversy, it may be worth checking this. Line 85 missing a close bracket. Need to clarify if any difference between school-based and tertiary education contexts of interventions and is unclear from this if school-based studies will be included. Objective 3. It may be preferable to state that common practice elements will be described in full, and then those identified as having positive effects, no effects, negative effects, if possible.	Thank you this sentence has now been shortened and clarified, line 78. Thank you, this clarification has been made, line 84. We have also removed 'resilience', line 91. Corrected. Thank you this has been clarified, line 100,107.

The population may not always be 18-30 so maybe not helpful to be so specific in terms of age. Identifying search terms What about 'psychoeducation'?	Thank you for this suggestion we have revised the objective, line 176. In line with reviewer 2 comment, we have removed the age range. Thank you for this, which has been added, line 222.
There is no mention of LMIC in searches that may result in exceptionally high number of returns. It is recommended a librarian is contacted as part of this review process for assistance on how to develop search strategy to capture those studies. There is greater specificity in terms of mental health literacy (self-care, resilience etc) but not in terms of mental health outcomes (anxiety, depression etc). If it is a small pool of studies further extending to these concepts may capture more of the interventions you wish to review. There could be more details provided about inclusion criteria for types of grey literature. There needs to be clearer emphasis earlier in the manuscript that you are only interested in in-person delivery and why. It is mentioned above but just needs to be stated more clearly.	Thank you for this suggestion, we have added the need to maintain feasibility of the number of records, and note the recommendation on involvement of the librarian, line 230. Thank you for this suggestion, we have now included outcomes in the searches, Line 225. We have included 'Programme reports, case studies and intervention manuals will be included', line 241.

Make a statement as to why formal assessment bias is not appropriate for this context. TIDier is the appropriate tool. Will the summarising of findings be a reiterative process?	Thank you, we have added 'hence in-person group approaches are the focus of this review, which will inform development of in-person interventions', line 145. We have added 'as the aim is to provide an overview of existing evidence regardless of methodological quality.', line 289. We would not consider the process of summarizing findings to be an iterative process, once articles have been identified the findings of each will be outlined.
Discussion Social determinants are mentioned but not part of the scoping protocol or research objectives. Will variation in country of origin take into account relevant social deprivation indices, for example. This may have relevance to implementation of any key recommendations and findings.	We had not planned to take into consideration social deprivation indices, as this could further complicate the review, however the suggestion is well noted, and we have made the change in the methods to allow for noting differences between high, low-income and lower middle income countries. '... and categorization according to low-income or lower-middle income country status (given differences in availability of resources across these settings).' Line 305.

Reviewer 2 comment	Author response
1) What is mental health promotion? Perhaps add a definition of mental health promotion in paragraph starting at line 75.	The meaning of mental health promotion has been clarified line 82.
2) Line 85: missing bracket after references 23, 27, 28	Corrected, thank you
3) Higher education or tertiary education or	Thank you for the comment, we have changed to

university – in UK, these all mean the same thing, but in the United States, there are more nuanced definitions. Clarify what you mean by the terms at the start and outline which wording you'll use throughout, or just use one. Either way, define the setting.	higher education throughout and included explanation, line 66.
4) This is a very long sentence to read. Perhaps break up so that it doesn't lose the important points it is making: Given that many mental 98 health conditions have their onset in adolescence this is a key point for intervention and there is 99 developing evidence that school-based prevention programmes can be effective in improving 100 mental health literacy and reducing mental health stigma defined as attitudes and beliefs regarding 101 mental disorders [25-28].	Thank you we have separated into two shorter sentences, from line 105.
5) Split this long sentence into two for strong reading and so that the paragraph has at least 3 sentences: Systematic reviews of prevention programmes for student 105 mental health showed moderate effects for common practice elements including psychoeducation, 106 relaxation techniques, and cognitive restructuring [19] as well as guided mental health skills 107 training programmes [22, 29-31] and computer and web-based interventions delivered by a variety 108 of professionals [32-34]	We have reworded to 3 sentences, Line 115.
6) Paragraph starting from line 111: Why have you chosen to focus on psychoeducation when you've listed other prevention programmes (i.e. relaxation techniques, cognitive restructuring, etc.) in the paragraph before that showed moderate effects?	Thank you, agreed that this focus on psychoeducation did not fit with the previous paragraph so have removed it.
7) Line 131, it may be useful to state why in-person interventions 'contribute to a different niche'	We have reworded this, and explained that in person interventions are important for support and rapport, and in settings where access to technology may be limited Line 142.
8) Line 141, 'These approaches have the potential to be low resource....' Some research argues that peer-led approaches are not 'cheap' (Turner, G. Peer support and young people's health. J. Adolesc. 1999, 22, 567–572.) I found something similar when speaking to staff who run peer programmes in higher education settings. Staff required more resources and capacity than they had to run the programmes effectively, which you may want address briefly. See my recent publication: Pointon-Haas J, Byrom N, Foster J,	Thank you for this interesting paper which we have now referenced in this section, line 155 and we have noted that peer approaches still require resources and engagement for effective implementation.

Hayes C, Oates J. Staff Perspectives: Defining the Types, Challenges and Lessons Learnt of University Peer Support for Student Mental Health and Wellbeing. Education Sciences. 2023; 13(9):962. https://doi.org/10.3390/educsci13090962	
9) Line 145: ‘in education and training settings’ – once again, clarify settings at start, this is first time you’ve used term ‘training settings.’	Clarified to ‘higher education settings’ to be consistent.
10) Research question 1: add ‘in-person’ as you have outlined in line 144. Also, with terms, in-person, group-based and peer-facilitated, use the hyphen (or don’t) consistently.	This has been changed throughout.
11) Line 186: The chosen age range (18-30) may limit a full picture for this review, as tertiary education students can be outside this age range, while I agree that the majority are not. Also, some studies may not report on age, so will you exclude these? Or, their age range may not match yours, so will you exclude these? It may be easier to just say ‘tertiary education students’ and remove the age range altogether.	Thank you for this suggestion, we have removed the age range.
12) Line 201: you may also want to consider using the educational database, ERIC (Education Resources Information Center)	Thank you for this suggestion which we have included, line 218.
13) Line 211: you can use term ‘backward citation tracking’ to describe looking at reference list of included studies.	Thank you for the suggestion, included in line 233.
14) Line 217: perhaps consider using OpenGrey and Grey Matters and cite the grey literature searches.	Thank you for the suggestion, included in line 240.
15) For searching, perhaps you can use existing relevant research to help you define your search terms. For example, in my recent systematic review, I used: John NM, Page O, Martin SC, Whittaker P. Impact of peer support on student mental wellbeing: a systematic review. MedEdPublish 2018; 7(3): 170. The search strategy I developed in my recent systematic review may help you decide an exhaustive search term list: Pointon-Haas J, Waqar L, Upsher R, Foster J, Byrom N, Oates J. A systematic review of peer support interventions for student mental health and well-being in higher education. BJPsych Open. 2023 Dec 15;10(1):e12. doi: 10.1192/bjpo.2023.603. PMID: 38098123; PMCID:	Thank you for this suggestion, we have now referenced your review, and added some of the terms that were used, Line 220 onwards.

PMC10755562.	
16) Line 218: while it's for systematic reviews to do independent screening, which may not be relevant for the scoping review, I found Rayyan helpful to use for screening after downloading citations into Endnote in a recent review: https://www.rayyan.ai/	Thank you for this suggestion, added in line 236.
17) Line 227, Inclusion Criteria: you may want to say why you have chosen the range of years '2007 to 2023' and also say when you are planning to undertake this search. Also, the impression I had reading this was that the review would only be of LMIC campuses, but you've included HIC as well in the inclusion criteria. Perhaps clarify this.	This date range has been explained Line 260 (for feasibility in the number of records to review and to bring out recent developments in the field). Thank you for this, we have included an explanation that HIC studies will be included as there may be elements and approaches that are relevant for LMIC where the field is developing, line 255.
18) Line 228: I'm assuming that you'll include any study internationally? Perhaps it would be useful to clarify that this is a global scoping review somewhere. Also, some studies do not report on the country where the study took place, I assume you will exclude those? It may be good to elucidate this in exclusion criteria.	'Global' has been added, line 159. We will still include studies that don't report the country as we may exclude some important learnings if we don't (this has been clarified, line 262).
19) Line 243, I think you're missing a word? 'Data extraction from...will be'	Thank you, corrected line 273.
20) Line 250: perhaps it would be good to add how the study defines a 'peer' into your fields. How a 'peer' is defined beyond being a student is interesting to see, since clinical settings define peer support workers as people with lived experience of mental health difficulties who use those experiences to share and help others. In contrast, tertiary education settings tend to bring students together based on the courses they share or their year of study. It may be interesting to gather this data to see how it is defined in studies.	Thank you for this important comment, we have added it to the charting approach, line 282.
21) Line 250: perhaps consider adding the number of sessions that took place in addition to the duration of the intervention. Many peer	Thank you, this addition is made line 282.

programmes have a set number of sessions that take place, and they don't report on duration.	
22) Line 250: In addition to sample size, you may want to report on retention if the programme happens over a period of time. Also, will you report on demographics of the peer leading the sessions and/or those attending the peer group if provided	Thank you, this addition is made line 282.
23) Line 250: How will you report on the 'positive effects on mental health' as your 3rd research objective outlines you will do? Will there be any sort of meta-analysis of statistics presented to consider effectiveness? In the 'summarizing and reporting findings' section, you may want to outline that you don't expect to be able to do a meta-analysis as the measures used have been reported to be heterogenous (see Pointon-Haas systematic review provided in point 15).	Thank you for this question. We will not be looking at effectiveness in terms of mental health outcomes. The objective has been reworded to 'To describe common practice elements and those identified as having positive effects, no effects, negative effects, if possible.' Line 176.
24) Line 250: perhaps you also want to add details on the training that the peers leading programme received, or the professional support they were given. These are the areas where costs are incurred, which make peer programmes less economical than sometimes assumed. (see point 8)	Thank you, included line 282.
25) Line 258: how will you report on your assessment of completeness? Will this be charted with the other data?	This will be reported on in the data charting, added to line 294.
26) Line 292: it may also be useful to say how big the population of students in tertiary education are to give an idea of scope.	We have now indicated the importance of youth in the global population, line 327.

VERSION 2 – REVIEW

REVIEWER	Judith Lunn Lancaster University
REVIEW RETURNED	04-Apr-2024

GENERAL COMMENTS	Thank you for the opportunity to re-review this interesting manuscript. The authors have satisfactorily addressed all of my comments in the tabulated Author Response attached document.
--